# Acoustic Localization of Bragg Peak Proton Beams for Hadrontherapy Monitoring [note 1]

**DOI:** 10.3390/s19091971

**Published:** 2019-04-26

**Authors:** Jorge Otero, Ivan Felis, Miguel Ardid, Alicia Herrero

**Affiliations:** 1Institut d’Investigació per a la Gestió Integrada de les Zones Costaneres (IGIC), Universitat Politècnica de València (UPV), Gandia, 46730 València, Spain; mardid@fis.upv.es; 2Centro Tecnológico Naval y del Mar (CTN), Fuente Álamo, 30320 Murcia, Spain; ivanfelis@ctnaval.com; 3Institut de Matemàtica Multidisciplinar, Universitat Politècnica de València (UPV), 46022 València, Spain; aherrero@mat.upv.es

**Keywords:** hadrontherapy, acoustic localization, Bragg peak, thermoacoustic, piezoelectric ceramic

## Abstract

Hadrontherapy makes it possible to deliver high doses of energy to cancerous tumors by using the large energy deposition in the Bragg-peak. However, uncertainties in the patient positioning and/or in the anatomical parameters can cause distortions in the calculation of the dose distribution. In order to maximize the effectiveness of heavy particle treatments, an accurate monitoring system of the deposited dose depending on the energy, beam time, and spot size is necessary. The localized deposition of this energy leads to the generation of a thermoacoustic pulse that can be detected using acoustic technologies. This article presents different experimental and simulation studies of the acoustic localization of thermoacoustic pulses captured with a set of sensors around the sample. In addition, numerical simulations have been done where thermo-acoustic pulses are emitted for the specific case of a proton beam of 100 MeV.

## 1. Introduction

Localization of a source is a technique in which a source is located by detecting propagated signals received in several sensors and the analysis of them [1]. There are many localization techniques proposed for wireless sensor networks [2,3]. However, in this article, a three-dimensional localization to solve the estimation of an acoustic source in a homogeneous medium is introduced. The use of acoustic sensors to locate sound sources in such practical systems is of great interest but needs further development and improved performance systems. This research has significant potential for many applications in medicine, physics, engineering, and underwater acoustics. The method to locate the tumor tissue is based on a computed tomography scan to find the area that will then be radiated by heavy particles in the Bragg peak region [4]. However, uncertainties in the patient positioning and/or in the anatomical parameters can increase the uncertainty during the radiotherapy. In these cases, acoustic source localization in medical applications has gained a lot of interest in recent years, which ought to be the necessity for improving the monitoring of tumor tissue in hadrontherapy treatments. Linear sensors can be employed for acoustic source localization in a noise environment using a time delay estimation. The method presented in this paper is based on the TDOA (time difference of arrival) [5] technique that performs very well in the localization of an acoustic event in both two-dimensional and three-dimensional spaces decreasing the error while increasing the number of sensors. The acoustic signal is generated and detected by piezoelectric sensors in known positions and using a DAQ system to record the signal. Differences in the signal propagation path from the source lead to different phases in the detected signal. Therefore, cross-correlation analysis is used to estimate the delays of arrivals accurately [6], even in conditions with low signal-to-noise ratio.

The pressure source localization of the Bragg peak in hadrontherapy can also be used to identify the regions of local heat released due to energy deposition. In this paper, we focus on the objective of monitoring the position for hadrontherapy through the Bragg proton beam acoustic localization. This pressure is related to the beam energy, the temporal pulse width, the size of the beam, and the number of protons by pulse, so, to some extent, it might be used in the future for dose sensitivity as well, but this aspect is out of the scope of the paper. For this reason, as a first approach, the source assessed in this article presents a pressure above the threshold of detection [7] for beams of a few million protons per spill of energies from 20 up to 200 MeV. Both in simulation and experiment, homogenous and isotropic medium is used and the wall effect is neglected since the direct signal arrives earlier to the Omni-directional receiver than the reflected signal.

## 2. Overview of Approach

Techniques based on cross-correlation and generalized correlation (GCC) [8] have been employed to determine the time difference of arrival of the signals (TDOA) given its computational cost and accuracy of the results. To obtain a better estimate of the TDOA, it is convenient to filter the signal before its integration, as shown in Figure 1 [6].

The cross-correlation $R_{x_{i}x_{j}}$ between the signal $x_{i}$ and $x_{j}$ filtered by the filters $H_{i}$ and $H_{j}$, is expressed as a function of the power spectral density $G_{x_{i}x_{j}}$, as shown below.
(1)$$R_{x_{i}x_{j}}^{GCC}\left(t^{\prime }\right)=\int_{-\infty }^{+\infty }H_{i}\left(f\right)H_{j}^{*}\left(f\right)G_{x_{i}x_{j}}\left(f\right)e^{i2\pi ft^{\prime }}df=\int_{-\infty }^{+\infty }\varphi^{GCC}\left(f\right)G_{x_{i}x_{j}}\left(f\right)e^{i2\pi ft^{\prime }}df$$
where {*} indicates a conjugated complex and $\varphi^{GCC}\left(f\right)$ is a frequency-dependent weight function. Due to finite observations, we can only obtain an estimation of $G_{x_{i}x_{j}}\left(f\right)$ [9]. Therefore, to obtain the TDOA, the following expression will be used [6].
(2)$${\hat{R}}_{x_{i}x_{j}}^{GCC1}\left(t^{\prime }\right)=\int_{-\infty }^{+\infty }\varphi^{GCC}\left(f\right){\hat{G}}_{x_{i}x_{j}}\left(f\right)e^{i2\pi ft^{\prime }}df$$
where ${\hat{G}}_{x_{i}x_{j}}\left(f\right)$ is the obtained estimation of $G_{x_{i}x_{j}}\left(f\right)$. For each pair of sensors, the TDOA is taken as the time delay that maximizes the cross-correlation between the filtered signals of both sensors, that is: ${\hat{\tau }}_{ij}^{GCC}=\arg\left(\max_{t^{\prime }}\left\{{\hat{R}}_{x_{i}x_{j}}^{GCC1}\left(t^{\prime }\right)\right\}\right).$

A general model for three-dimensional (3-D) estimation of a source using M receivers is developed. To obtain the location of the source, we start by knowing the spatial position $\left(x_{i},y_{i},z_{i}\right)$ of a certain number of sensors. Let $\left(x_{s},\;y_{s},\;z_{s},\right)$, the position of the source to be located, the distance between the source and the *i*-th sensor will be:(3)$$d_{i}=\sqrt{{\left(x_{i}-x_{s}\right)}^{2}+{\left(y_{i}-x_{s}\right)}^{2}+{\left(z_{i}-x_{s}\right)}^{2}}$$

The range difference in distance between the *i*-th receiver and the first receiver, $d_{i1}$, is given by:(4)$$d_{i1}=c\cdot \tau_{i1}=d_{i}-d_{1}=\sqrt{{\left(x_{i}-x_{s}\right)}^{2}+{\left(y_{i}-y_{s}\right)}^{2}+{\left(z_{i}-z_{s}\right)}^{2}}-\sqrt{{\left(x_{1}-x_{s}\right)}^{2}+{\left(y_{1}-y_{s}\right)}^{2}+{\left(z_{1}-z_{s}\right)}^{2}}$$
where c is the sound velocity in the medium, $d_{1}$ is the distance between the first receiver and the source, and $\tau_{i1}$ is the estimated TDOA between the first receiver and the *i*-th receiver [10].

Equation (4) considered for all the sensors form a nonlinear equation system whose solution can be found by several ways. After studying different resolution methods, it was decided to use the Newton-Raphson method since it offers very good results and computation time.

### Newton-Raphson Method

To get the position of the thermoacoustic source inside the medium, we have solved the nonlinear system using the Newton-Raphson method [11] by means of partial derivatives. Consider a system of m equations and n unknowns.

(5)$$f_{m}\left(x_{1},x_{2},x_{3},\ldots ,x_{n}\right)=0$$

This system can be written in vector form as f(x)=0, where f is a vector of m dimensions and x is a vector of n dimensions. To solve this system of equations, we have to find a vector x such that the function f(x) equals the null vector. If we call η  to the solution of the system and $x_{r}$ to an approximation of it, we can develop f in Taylor series around this approximation as:(6)$$f\left(x\right)=f\left(x_{r}\right)+\nabla f\left(x_{r}\right)\cdot \left(x-x_{r}\right)+\cdots$$

Since f(η)=0 then we get, as an approximation, the following.

(7)$$0\approx f\left(x_{r}\right)+\nabla f\left(x_{r}\right)\cdot \left(\eta -x_{r}\right)$$

Now, we can define the vector $x_{r+1}$ as this approximation, which is closer to the root than $x_{r}$. We can continue with the iterative method to obtain approximations closer and closer to the solution. To write the iterative method, the term $\nabla f\left(x_{r}\right)$ is replaced by the Jacobian of the function f, that is:(8)$$J(x_{r})=[\begin{array}{cccc} \frac{\partial f_{1}}{\partial x_{1}} & \frac{\partial f_{1}}{\partial x_{2}} & \cdots & \frac{\partial f_{1}}{\partial x_{n}}\\ \frac{\partial f_{2}}{\partial x_{1}} & \frac{\partial f_{2}}{\partial x_{2}} & \cdots & \frac{\partial f_{2}}{\partial x_{n}}\\ \vdots & \vdots & \vdots & \vdots \\ \frac{\partial f_{m}}{\partial x_{1}} & \frac{\partial f_{m}}{\partial x_{2}} & \cdots & \frac{\partial f_{m}}{\partial x_{n}} \end{array}]$$
which is a n × m matrix. Then, it is possible to obtain a new value of $x_{r+1}$ by solving the following relationship.

(9)$$J(x_{r})\left(x_{r+1}-x_{r}\right)=f\left(x_{r}\right)$$

Then, iteratively, we can approximate more and more the $x_{r+1}$ to η until a solution error $\left|x_{r}-x_{r+1}\right|$ previously fixed is reached.

This method has been compared with other algorithms for solving systems of nonlinear equations and has shown some advantages over the rest like good accuracy and low computing cost. However, if the initial solution value of the system differs greatly from the real solution, then the method does not converge conveniently. Figure 2 shows the convergence of the location algorithm as a function of the distances between the initial point of the method, the result of the reconstruction of the position and the real position for a total of 10,000 simulations.

## 3. Thermoacoustic Simulation

### 3.1. Bragg Peak

The main physical advantage of heavy particles as compared to photons is their characteristic depth-dose profile, the known Bragg curve in honor of Sir William Henry Bragg who investigated the energy deposition of alpha particles, which form a Radium source in the air at the beginning of the last century [12]. While the photon dose decreases exponentially with penetration depth according to the absorption law for electromagnetic radiation, the depth-dose of heavy charged particles exhibits a flat plateau region with a low range of the particles [13]. This paper uses an analytical approach presented by T. Bortfiel in 1996 [4], and the numerical representation valid for protons with energies between 10 MeV and 200 MeV. Thus, the energy of single protons along a Z-axis in a homogeneous medium (water) is considered. The total energy released in the medium per unit mass in the Z-axis is shown below.
(10)$$T\left(z\right)=-\frac{1}{\rho }\left(\phi \left(z\right)\frac{dE\left(z\right)}{dz}+\frac{d\phi \left(z\right)}{dz}E\left(z\right)\right)$$
where ϕ(z) is the proton flow, that is, the number of protons per $cm^{2}$, E(z) is the energy deposited on the Z-axis, and ρ represents the mass density of the medium. The method makes use of a midpoint where a certain fraction γ of the energy released in nuclear interactions is absorbed locally while the rest is ignored. Then, the total absorbed energy $\hat{D}\left(z\right)$, will be given by:(11)$$\hat{D}\left(z\right)=\;-\frac{1}{\rho }\left(\phi \left(z\right)\frac{dE\left(z\right)}{dz}+\gamma \frac{d\phi \left(z\right)}{dz}E\left(z\right)\right)$$

A relationship between the initial energy $E\left(z=0\right)=E_{0}$ and the range $z=R_{0}$ in the medium can be approximated as $R_{0}=\alpha E_{0}^{p}$ for p=1.5. This relation is valid for protons with energy close to 250 MeV. The factor α is proportional to the square root of the effective atomic mass of the medium [4]. Using the inverse for $R_{o}\leq 0.5\;\mathrm{cm}$ and assuming $E_{o}$ to be given in units of MeV, the best fit parameters for $E_{o}(R_{o})$ are p=1.77, $\alpha =2.2\times 10^{-3}$ for the proton in water [4]. The remaining energy E(z) at an arbitrary depth $z\leq R_{o}$ fails to travel the distance $R_{o}-z$ according to the range-energy relationship shown below.

(12)$$E\left(z\right)=\frac{1}{\alpha^{1/p}}{\left(R_{0}-z\right)}^{1/p}$$

For energies above 20 MeV, there is non-negligible probability that protons may be lost from the beam due to nuclear interactions. This non-elasticity was studied and tabulated by Janni [14] as a function of the residual range $\left(R_{0}-z\right)$. The proton flow ϕ(z) can be written by using the equation below.
(13)$$\phi \left(z\right)=\phi_{0}\frac{\beta }{1+\beta R_{0}}$$
where $\phi_{0}$ is the primary fluence and the slope parameter β was determined to be $\beta =0.012\;{\mathrm{cm}}^{-1}$ [4]. Thus, the distribution of the deposition of energy along the depth range can be expressed as the equation below.
(14)D(z)= Φ0eζ2/4σ1pΓ(1p)2πρpα1p(1+βR0)×[1 σL−1/p(−ζ)+(βp+γβ+εR0)L−1/p−1(−ζ)]
where Γ represents the gamma function, $\zeta =\left(R_{0}-z\right)/\sigma$, with a σ value of $0.012R_{0}^{0.935}$, and ε represents a relatively small fraction of the fluence $\Phi_{0}$ in the peak. Figure 3 shows the distribution of the dose as a function of the range for a different proton energy.

### 3.2. Thermoacoustic Model

In the thermoacoustic case, an excited point source emits a pressure wave proportional to the first time derivate of the excitation pulse [15]. Hence, a Gaussian excitation pulse leads to a bipolar acoustic emission consisting of a positive compression, which results in an increase in pressure. This is followed by a negative rarefaction, which is a decrease of pressure. The positive and negative pressure peaks are not only due to the heating and cooling of the medium, but the variation of the heating rate also plays a role. The medium expands or contracts according to its coefficient of thermal volumetric expansion α’. As a result, a pressure wave is observed. The pressure wave from an energy deposition in a region can be understood as the sum of the individual responses that would be observed from decomposing the spatial deposition into point sources. The resulting pressure signal depends on the time derivative of the excitation pulse. The amplitude of the wave depends on the energy deposited, the number of protons per pulse of the beam, and the temporal shape of the excitation pulse. A dose of 1 Gy generates a ~240 μK temperature increase in water [15]. Ignoring heat diffusion and cinematic viscosity, the wave equation that describes the pressure p at a time t and position $\vec{r}$, is shown below [16,17,18,19].
(15)$${\bar{\nabla }}^{2}p\left(\vec{r},t\right)-\frac{1}{c_{s}^{2}}\cdot \frac{\partial^{2}p\left(\vec{r},t\right)}{\partial t^{2}}=-\frac{\alpha^{\prime }}{C_{p}}\cdot \frac{\partial^{2}\epsilon \left(\vec{r},t\right)}{\partial t^{2}}$$
where $c_{s}$ (ms^−1^) represents the speed of sound in the middle, $C_{p}\;$ (${J\;\mathrm{kg}}^{-1}K^{-1}$) is the specific heat capacity, and $\epsilon \left(\vec{r},t\right)\;$(${J\;s}^{-1}m^{-3}$) is the energy density deposited in the medium. Equation (15) can be solved using the Kirchhoff integral as shown below.
(16)$$p\left(\vec{r},t\right)=\frac{1}{4\pi }\frac{\alpha^{\prime }}{C_{p}}\int_{V}^{}\frac{dV^{\prime }}{\left|\vec{r}-{\vec{r}}^{\prime }\right|}\cdot \frac{\partial^{2}}{\partial t^{2}}\epsilon \left({\vec{r}}^{\prime },t-\frac{\left|\vec{r}-{\vec{r}}^{\prime }\right|}{c_{s}}\right)$$
where $p\left(\vec{r},t\right)$ denotes the hydrodynamic pressure at a given place and time. The values for the thermoacoustic model were an energy of 100 MeV, a temporal profile of 1 μs, $3.4\times 10^{6}$ protons per pulse, the beam with a size of 1 mm, and a sensor located 40 mm from the Bragg peak. The characteristics of the simulation are given by simulation results from different studies, as well as their application in clinical cases [7,15,20,21,22,23,24,25]. The values for this case are shown in Figure 4. As a result, the pressure obtained at the reception point will be the signal that will be emitted by the piezoelectric [26] transducer to simulate a bipolar source that will be located by the sensor array.

## 4. Experimental Setup

The experimental data measurements were made in the laboratories of the physics department at the Universitat Politècnica de València (Spain). There is a water tank with a volume of $0.64{\;m}^{3}$ with a programmable 3D axis system MOCO PI MICOS arm that was programmed to move the Reson TC4014 receiver hydrophone in the tank. The hydrophone has a receiving sensitivity of −186±3 dB @1 V/μPa and a frequency response from 15 kHz to 480 kHz. The emitter hydrophone is a Reson TC4038 with a transmitting response of 110 dB @1 μPa/V @ 1m and a frequency response from 50 kHz to 800 kHz. Figure 5 shows the experimental setup with the transmitter and receiver inside the tank. A National Instruments data acquisition system was used with PXI type cards to generate the signal used as input of the linear E&I A150 amplifier that feeds the transmitter. Both the receiving and the feeding signal were captured. The latter was captured with an ×100 probe to avoid overloads in the system. All signals were stored at 10 Ms/s with a duration of 500 µs.

Two experiments have been performed. First, a calibration is done to reduce the uncertainty due to the time of arrival from which the speed of the sound has been measured. For this, 12 different reception positions were assessed in a straight line on the emitter axis. By having the 12 measurements along the line, a time-distance linear adjustment is made whose slope value corresponds to the speed of sound. In addition, the calibration allows a time correction that results in a decrease of the arrival time of the signal, according to the linear adjustment obtained in the fit.

In the second part of the experiment, 12 reception points have been set that will correspond to the 12 sensor positions of the array. The mechanical axis moves the hydrophone to each of the points and then the signal is emitted and recorded 10 times. Figure 5b shows one of the measuring points where the Reson TC4014 sensor is fixed to the mechanical arm.

## 5. Studies and Results

### 5.1. Numerical Simulation

To evaluate the localization method described, the reconstruction of the location of a Gaussian pulse source of 50 μs is simulated from the reception of four sensors located on the lateral surface of different coordinates. Figure 6 shows the position of the sensors and the source in the space for the simulated model [6]. In this simulation, the sources are always into the volume covered by the coordinates of the sensors. Despite it being possible to locate the source with three sensors, the use of at least four sensors is incorporated to both improve the reliability and quality of the results.

To evaluate the algorithm, the volume of the cube has been modified between $27.0\times 10^{-3}\;m^{3}$ and $512.0\times 10^{-3}\;m^{3}$. Positions of the sensors are shown in Table 1, where H represents the length of the cube, whose values were 200, 300, 400, 500, and 600 mm [6].

Table 2 shows the deviation of the position of the simulated source with respect to the real position of the source. The results can be expressed as a function of the distance between the reconstructed position and the real position of the source. Figure 7 shows the results, where the abscissa axis shows the volume in $m^{3}$, while the ordinates axis shows the distance in mm between the prediction of the algorithm and the real position. In addition, a fitting line to the results is shown.

These reconstructed positions do not exceed 5 mm of the real position for all the studied cases. In addition, it is important to calculate the algorithm for a future application in real time. For this reason, Figure 8 shows the calculation times for 4, 6, 8, 10, and 12 sensors for a source position, depending on changes in volume.

Once satisfactory results of the localization algorithm have been obtained by simulating different known source points in known sensor positions, the localization method has been evaluated experimentally in the next section.

### 5.2. Experimental Localization with Thermoacoustic Signals

To validate the localization method, 12 reception positions (sensors) have been configured for a single source position. The signal emitted is the pulse obtained by the simulation of the thermoacoustic model, given by Equation (15). Table 3 shows the positions of the measuring points, which have all been referenced to the lower corner of the tank.

Two inputs channels were used. Channel 1 recorded the emitted signal passed to the amplification system while channel 2 recorded the received signal of the emitter Reson TC4014. Both signals were used afterward for correlation. The receiver is fixed to the MOCO programmed axis that moves the hydrophone to each of the measurement positions from position 1 to position 12 in the direction shown in Figure 9. By doing this, we have the same information as when we using a 12-element sensor array.

The analysis of the measurements has taken into account the calibration of the measurement of the speed of sound 1492 m/s. This correction allows for a better detection precision in the TOA. In addition, the speed of the localization algorithm and the accuracy of the results will be evaluated, according to the number of reception points. Figure 10 shows an example of the signal emitted and the signal received as well as the correlation between them in order to obtain the TOA as the time of maximum amplitude of the correlation [9].

Table 4 shows the estimated location of the source (mean and standard deviation) for different groups of sensors labelled, according to Figure 9 and Table 3. From Table 4, we observe that the results obtained with this configuration are satisfactory and below 1 mm. Furthermore, similar accurate results are also possible with fewer sensors such as four sensors surrounding the source and covering different directions isotropically. In cases that sensors are only in a specific region, larger offsets (several mm) may appear in specific coordinates.

## 6. Conclusions

In this paper, the proton range of the analytical medium has been used to calculate the distribution of the dose in space. Specifically, we have addressed values of time between 1 µs and 10 µs, energies between 20 MeV and 200 MeV, and protons per pulse from $3.1\times 10^{8}$ to $8\times 10^{8}$. In the same way, an analytical model to calculate the distribution of the initial pressure at a single point has been used with a computing program to find the numerical solution of the general wave equation and calculate the acoustic waves resulting from the initial pressure. Some simulation parameters include pulse widths, beam energy, and spatial and temporal configuration. Thus, it is possible to determine the parameters and frequency spectrum to select the frequency responses of the transmitter and the receiver for the experiment.

This study has described and tested a procedure for monitoring the location of a hadrontherapy acoustic source based on the detection of the signal through piezoelectric sensors and on a model for calculating the position of the energy deposition. The localization algorithm has been applied for different configurations of sensors. Thus, the results show a significant improvement when a greater number of sensors is used. For a minimal set of four sensors, the results are better if the sensors cover different directions of the space. The accuracy of the results improves as the number of sensors increases, as shown in Table 4. Although the calculation time increases with the number of sensors, the difference is not significant for any of the proposed cases. Thus, the results indicate that it would be possible to monitor in real time the hadrontherapy treatment acoustically. At first look, the case studied may be considered too simplistic since the human body is neither homogenous nor isotropic and sensors are not omnidirectional. Therefore, for a practical case in hadrontherapy, all these aspects should be considered when taking care of the geometry and tissues involved, the real response of sensors, etc. However, for most of the cases, the results in small differences in the technique proposed, and the situation considered in the paper is good, and most probably the best, for a first general assessment of the technique proposed.

## Figures and Tables

**Figure 1 sensors-19-01971-f001:**
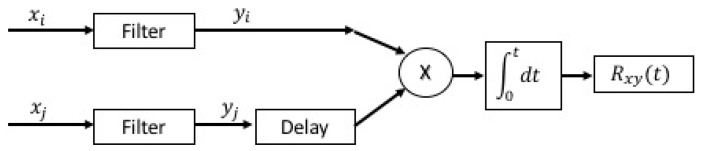
Scheme for obtaining the time of arrival (TOA).

**Figure 2 sensors-19-01971-f002:**
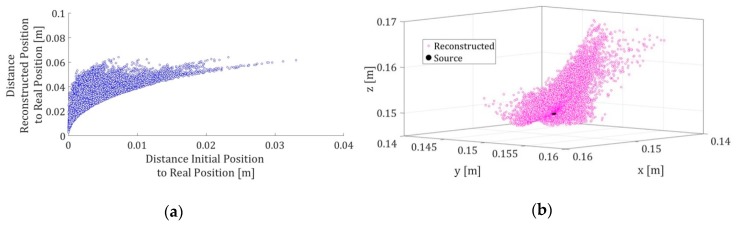
(**a**) The distance between the initial position for the algorithm and the real position of the source is shown on the abscissa axis, while the axis of the ordinates shows the distance between the signal reconstructed by the algorithm and the real position of the source; (**b**) The reconstructed positions for each of the 10,000 simulations are shown together with the real position (black).

**Figure 3 sensors-19-01971-f003:**
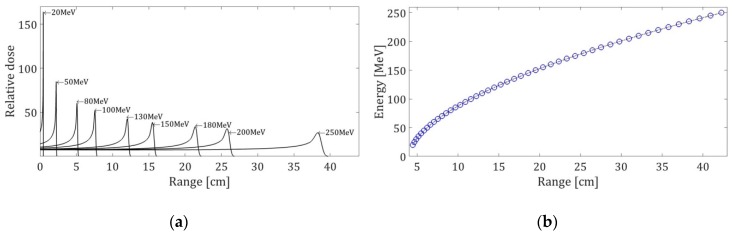
Bragg peak for different energies. (**a**) The deposition of the dose varies according to the energy of the proton. The maximum of the Bragg peak varies according to the energy; (**b**) The relationship Range–Energy for protons in water is shown.

**Figure 4 sensors-19-01971-f004:**
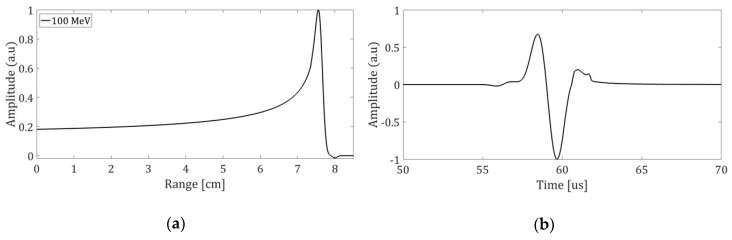
(**a**) Bragg curves with an initial energy of 100 MeV protons in water. The line represents the dose contribution from the fraction of protons that have nuclear interactions; (**b**) Pressure for a sensor located 4 cm from the Bragg peak on the axis of symmetry of the emission.

**Figure 5 sensors-19-01971-f005:**
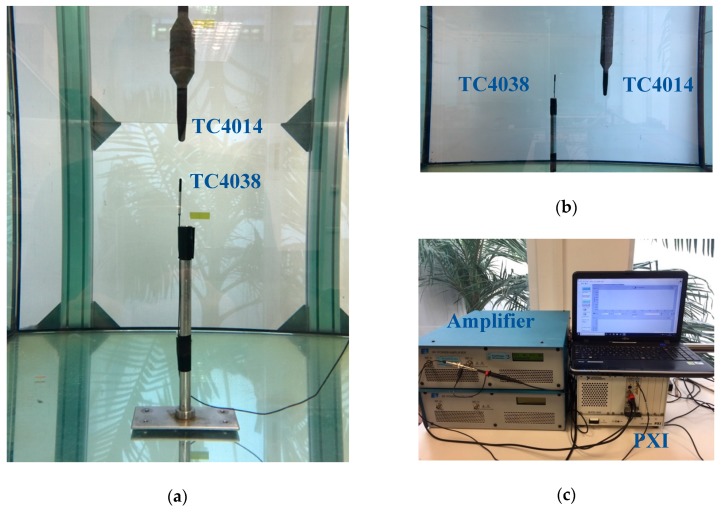
(**a**) The emitter and receiver are located as close as possible to each other to calibrate the motors; (**b**) The first position for measurements of sound speed and location; (**c**) System of generation and capture of signals.

**Figure 6 sensors-19-01971-f006:**
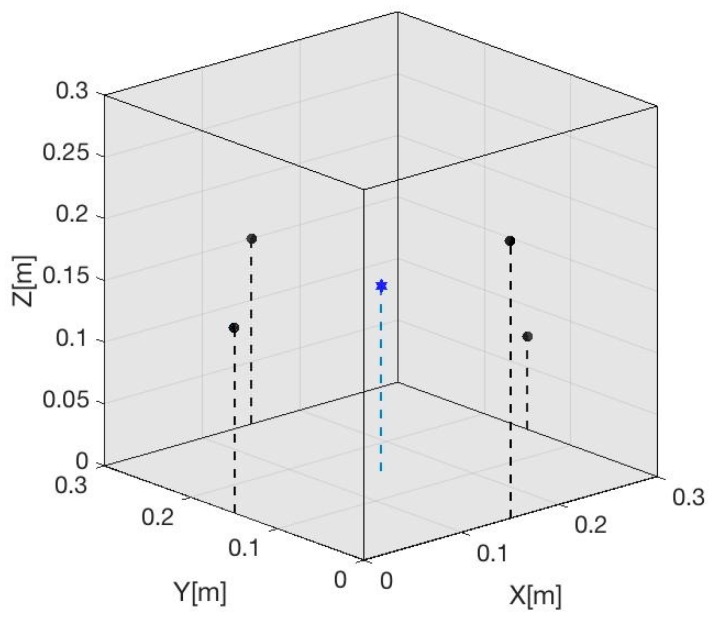
Volume proposed to evaluate the localization algorithm. In this case, four sensors (black points) have been situated on the sides of the cube. Inside, three events will be simulated in different positions. The positions of the sensors and sources are shown in Table 1. This figure shows the point source (blue point).

**Figure 7 sensors-19-01971-f007:**
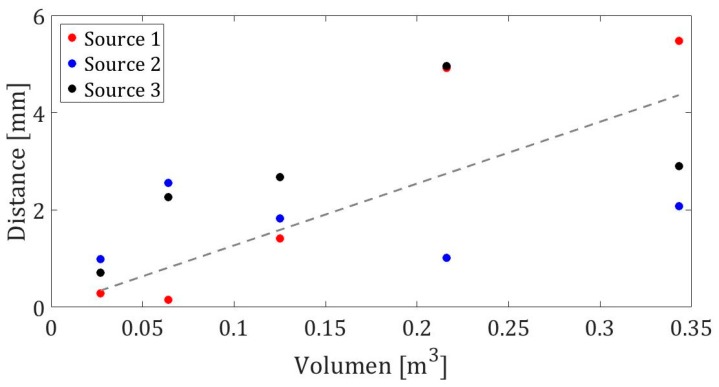
The positions of sources 1, 2, and 3 are shown in red, blue, and black, respectively. The dotted line is a fitting line to the results.

**Figure 8 sensors-19-01971-f008:**
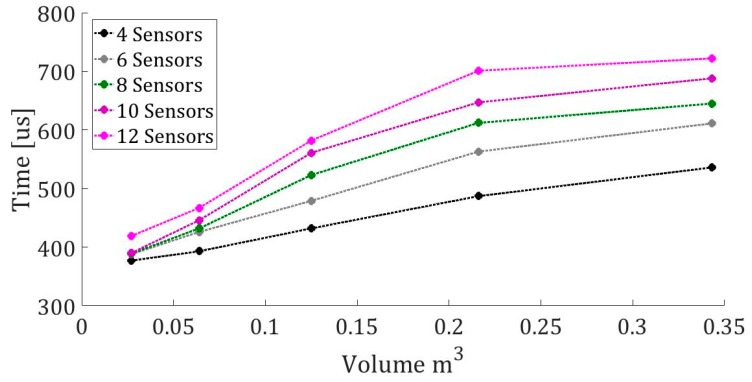
Calculation time with an Intel i5 processor for different sensors and volume.

**Figure 9 sensors-19-01971-f009:**
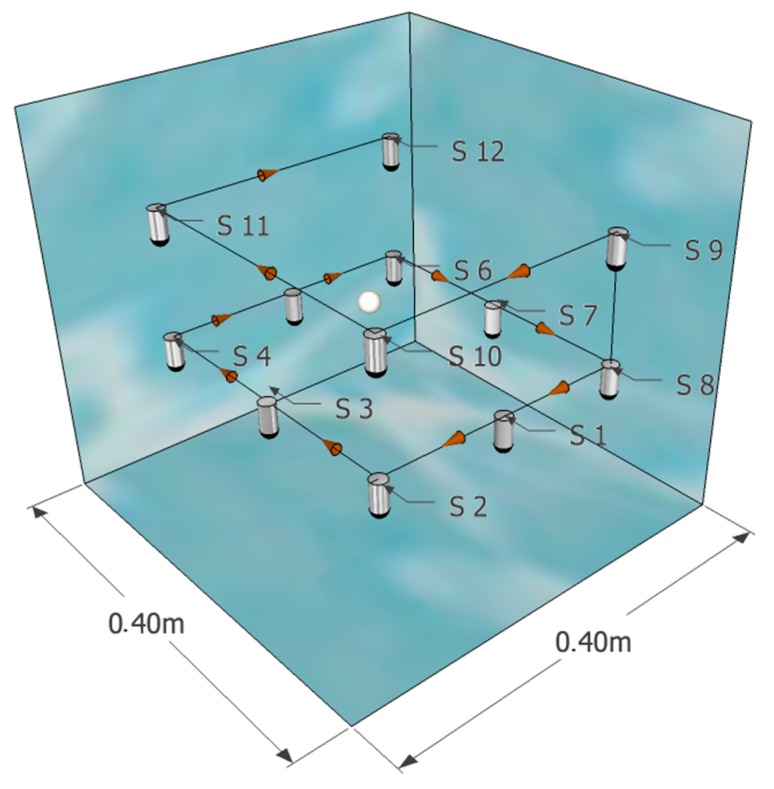
This figure shows the configuration of the receiver positions. The source is represented by a sphere (white) that is inside the volume generated by the sensors (cylinders of grey color with black tip) whose route is shown with the black lines and the conical red marks. In this figure, a smaller volume is represented inside the tank for its correct visualization.

**Figure 10 sensors-19-01971-f010:**
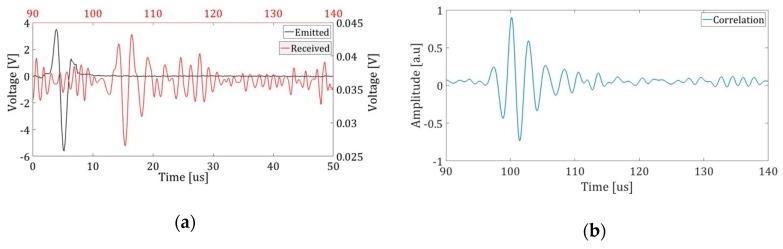
The figure shows the process of detecting the signal from the correlation between the emitted and received signal. (**a**) The signal generated in the simulation, shown in black, is emitted by the transmitter. The captured signal for point 1 is shown in red; (**b**) The arrival time can be extracted from the maximum value of the correlation of signals.

**Table 1 sensors-19-01971-t001:** Positions of the sensors and the source in the simulations.

Axis	Sensors				Source (mm)		
	1	2	3	4	1	2	3
X	H/2	0.0	H/2	H	100	100	80
Y	0.0	H/2	H	H/2	100	180	100
Z	3H/4	H/2	H/2	H/4	100	150	180

**Table 2 sensors-19-01971-t002:** Real and estimated positions of three different sources depending on the volume case.

Coord.	Real Position (mm)	Estimated Position (mm) for the Volume (m^3^)				
		$27\times 10^{-3}$	$64\times 10^{-3}$	$125\times 10^{-3}$	$216\times 10^{-3}$	$343\times 10^{-3}$
X	100	99.89 ± 0.07	100.03 ± 0.02	100.79 ± 0.55	101.97 ± 1.72	97.84 ± 1.50
Y	100	99.94 ± 0.04	100.02 ± 0.01	100.63 ± 0.45	102.26 ± 1.91	97.19 ± 2.02
Z	100	99.75 ± 0.17	99.85 ± 0.10	100.99 ± 0.70	103.90 ± 2.06	95.83 ± 2.91
X	100	100.63 ± 0.45	99.64 ± 0.03	100.19 ± 0.01	100.41 ± 0.01	98.94 ± 0.40
Y	180	179.45 ± 0.38	178.55 ± 0.10	180.68 ± 0.01	180.39 ± 0.01	179.44 ± 0.42
Z	150	150.52 ± 0.37	147.92 ± 1.55	151.68 ± 0.12	150.84 ± 0.01	148.31 ± 1.12
X	80	79.80 ± 0.13	80.87 ± 0.60	78.65 ± 1.02	78.43 ± 1.11	78.52 ± 1.01
Y	100	100.04 ± 0.03	101.04 ± 0.71	98.67 ± 0.92	96.33 ± 2.61	98.43 ± 1.10
Z	100	99.32 ± 0.47	101.81 ± 1.30	98.20 ± 1.32	97.05 ± 2.08	98.07 ± 1.42

**Table 3 sensors-19-01971-t003:** Position of the source and sensor reception points.

Axis	Sources Position	Sensor Positions [cm]											
		1	2	3	4	5	6	7	8	9	10	11	12
X	54.0	70.5	70.5	56.5	42.5	42.5	42.5	56.5	70.5	70.5	70.5	42.5	42.5
Y	53.0	53.0	40.5	40.5	40.5	53.0	65.0	65.0	65.0	65.0	40.5	40.5	65.0
Z	38.0	31.0	31.0	31.0	31.0	31.0	31.0	31.0	31.0	43.0	43.0	43.0	43.0

**Table 4 sensors-19-01971-t004:** Reconstructed position for different groups of sensors.

Group	Number of Sensors	Sensor	Estimated Location [cm]		
			X	Y	Z
1	4	2, 4, 6, 8	53.08 ± 0.87	53.10± 0.31	30.25± 0.72
2	4	9, 10, 11, 12	53.17 ± 0.18	53.12 ± 0.25	27.31 ± 0.21
3	4	6, 8, 9, 12	54.45 ± 1.11	60.04 ± 4.61	37.79 ± 2.02
4	4	2, 4, 10, 11	53.11 ± 2.10	53.05 ± 0.50	38.28 ± 0.41
5	4	3, 4, 9, 11	53.98 ± 0.44	53.01 ± 0.44	37.98 ± 0.70
6	4	1, 4, 9, 12	53.99 ± 0.20	53.03 ± 0.70	37.97 ± 1.21
7	6	2, 5, 7, 9, 11, 12	53.99 ± 0.10	53.01 ± 0.26	38.00 ± 0.20
8	8	1, 3, 4, 6, 7, 8, 10, 11	54.01 ± 0.24	53.01 ± 0.12	38.01 ± 0.35
9	10	1, 3, 4, 5, 7, 8, 9, 10, 11, 12	54.00 ± 0.17	53.00 ± 0.14	38.01 ± 0.24
10	12	All of them	54.01 ± 0.17	53.00 ± 0.12	37.99 ± 0.38

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
