# Peer review of "Acoustic Localization of Bragg Peak Proton Beams for Hadrontherapy Monitoring [Author-notes fn1-sensors-19-01971]"

_sensors, 2019, doi:10.3390/s19091971_

Reviewer 1 Report

The work is devoted to the actual topic. The correctness of the research methods and the novelty of the results are beyond doubt. It is desirable to include answers to the following questions in the text of the article:

1. What is the degree of truncation of the system of nonlinear equations in experimental studies? The possibility of real-time calculations is approved. What software? What is the performance of computing devices?

2. Are reflections from the cube walls taken into account in the experiment? Does this take into account the algorithm used?

3. The experiment was conducted in an water medium. The tissues of the human body are not all water-like in acoustic properties. How does the presence of acoustic impedance of these structures (for example, bone) affect the performance of the algorithm? How to filter thermoacoustic signals from these structures?

4. In the real situation on the human body, the use of spherical receiving hydrophones is difficult. The properties of other types of sensors will affect the shape of the received thermoacoustic signals. How does the calculation algorithm take into account this fact?

 It is desirable to include at least a few proposals on each of these issues.

Author Response

The answers have been added in an attached file and Ms. ALisa Zhang has another new version of the manuscript. 

Thanks

Reviewer 2 Report

The paper presented a Bragg Peak localization method with thermoacoustic signals based on cross-correlation method. Experimental and simulation studies were also presented to show the acoustic localization of thermoacoustic pulses captured with a set of sensors around the sample. Despite the importance of the study for hadrontherapy monitoring applications, I have several concerns below:

1. How can this method being applied for in vivo measurements? In clinic, the generated acoustic signal from the Bragg peak is not a point source. That should be discussed.

2. The resolution and dose sensitivity of measurements should be discussed.

3. The authors mentioned that generalized correlation method were used to determinate the time difference of arrival of the signals given its computational cost and accuracy of the results. However, this correlation will not be accurate when detecting weak thermoacoustic signals. One suggestion would be to use a lock-in amplifier for time-resolved photoacoustyic measurement. (Zhao, Yue, et al. "Simultaneous optical absorption and viscoelasticity imaging based on photoacoustic lock-in measurement." Optics letters 39.9 (2014); Zhao, Yue, et al. "Time-resolved photoacoustic measurement for evaluation of viscoelastic properties of biological tissues." Applied Physics Letters 109.20 (2016): 203702.) 

4. Table 4 shows the estimated location of the source with at least 4 sensors. Is 4 sensors the minimum number of sensors required to achieve this accuracy?

5. Detailed literature review is needed in the field of radiotherapy monitoring techniques:

1)Sarah  K. Patch  Daniel E.M. Hoff  Tyler B. Webb  Lee G. Sobotka  Tianyu Zhao. Two‐stage ionoacoustic  range verification leveraging Monte Carlo and acoustic simulations to stably account for tissue inhomogeneity and accelerator–specific  time structure  – A simulation study, Medical Physics, vol. 45, issue 2, pp. 783-793.

2)Sebastian  Lehrack, Walter Assmann, Damien Bertrand, Sebastien Henrotin, Joel  Herault, Vincent Heymans, Francois Vander Stappen, Peter G Thirolf,  Marie Vidal, Jarno Van de Walle and  Katia Parodi. Submillimeter ionoacoustic range determination for  protons in water at a clinical synchrocyclotron. Phys Med Biol. 2017 Aug  18;62(17):L20-L30.

3) Xiang, Liangzhong, et al. "X‐ray  acoustic computed tomography with pulsed x‐ray  beam from a medical linear accelerator." Medical physics 40.1 (2013): 010701.

4) Xiang, Liangzhong, et al. "High resolution X-ray-induced acoustic tomography." Scientific reports 6 (2016): 26118.

5)  Susannah Hickling, Hao Lei, Maritza A. Hobson, Pierre Thomas Léger,  Xueding Wang, Issam M. El Naqa. Experimental evaluation of x-ray  acoustic computed tomography for radiotherapy  dosimetry applications. ed. Phys. 44 (2), February 2017.

 Specific Comments:

1) "Table 1" (line 211) should be "Table 1.".

2) "27?3 64?3 125?3 216?3 343?3"  (line 212)  It should be "27 x 10-3 64 x 10-3 125 x 10-3 216 x 10-3 343 x 10-3"?

3) "Experimental lozalization with thermoacoustic signals" (line 225) the word "lozalization" is incorrect here. It should be "localization". Please proofreading the entire manuscript to eliminated such typos.

Author Response

(The authors gave the same response as above.)
